# Gene-drive suppression of mosquito populations in large cages as a bridge between lab and field

Andrew Hammond [1,2,9], Paola Pollegioni [3,4,9], Tania Persampieri[3,9], Ace North [5], Roxana Minuz[3], Alessandro Trusso[3], Alessandro Bucci[3], Kyros Kyrou [1], Ioanna Morianou[1], Alekos Simoni[1,3], Tony Nolan [1,6,10 ✉], Ruth Müller [3,7,8,10 ✉] & Andrea Crisanti[1,10 ✉]

CRISPR-based gene-drives targeting the gene *doublesex* in the malaria vector *Anopheles gambiae* effectively suppressed the reproductive capability of mosquito populations reared in small laboratory cages. To bridge the gap between laboratory and the field, this gene-drive technology must be challenged with vector ecology.

Here we report the suppressive activity of the gene-drive in age-structured *An. gambiae* populations in large indoor cages that permit complex feeding and reproductive behaviours. The gene-drive element spreads rapidly through the populations, fully supresses the population within one year and without selecting for resistance to the gene drive. Approximate Bayesian computation allowed retrospective inference of life-history parameters from the large cages and a more accurate prediction of gene-drive behaviour under more ecologically-relevant settings.

Generating data to bridge laboratory and field studies for invasive technologies is challenging. Our study represents a paradigm for the stepwise and sound development of vector control tools based on gene-drive.

[1] Department of Life Sciences, Imperial College London, London, UK. [2] Department of Molecular Microbiology and Immunology, Johns Hopkins Bloomberg School of Public Health, Johns Hopkins University, Baltimore, MD, USA. [3] Genetics & Ecology Platform, Polo d'Innovazione di Genomica Genetica e Biologia, Terni, Italy. [4] National Research Council Research Institute on Terrestrial Ecosystems, Terni, Porano, Italy. [5] Department of Zoology, University of Oxford, Oxford, UK. [6] Department of Vector Biology, Liverpool School of Tropical Medicine, Liverpool, UK. [7] Institute of Occupational, Social and Environmental Medicine, Goethe University, Frankfurt am Main, Germany. [8] Department of Biomedical Sciences, Institute of Tropical Medicine Antwerp, Antwerp, Belgium. [9] These authors contributed equally: Andrew Hammond, Paola Pollegioni, Tania Persampieri. [10] These authors jointly supervised this work: Tony Nolan, Ruth Müller, Andrea Crisanti. ✉email: Tony.Nolan@lstmed.ac.uk; rmuller@itg.be; a.drcrisanti@imperial.ac.uk

CRISPR-based gene drives are selfish genetic elements that can be used to modify entire populations of the malaria mosquito for sustainable vector control. First proposed in 2003, these elements use a mechanism of cut and paste (homing) in the germline to facilitate their autonomous spread from a very low initial release frequency[1,2]. One potentially powerful strategy aims to reduce the total number of mosquitoes by spreading a mutation that blocks female reproduction[3]. To be effective for the control of malaria in sub-Saharan Africa, such a strain must be able to compete effectively with wild populations of *Anopheles gambiae*, and remain effective over the medium to long-term. To this end, we and others have adopted a step-wise approach to the development and testing of gene drives in progressively rigorous and challenging conditions[4].

First generation suppression drives failed to maintain their spread when tested in small, caged-population experiments within an Arthropod Containment Level 2 laboratory because of the creation and selection of drive-resistant alleles, sometimes exacerbated by unintended fitness costs in 'carrier' individuals[5–8]. One strategy to mitigate against the likelihood of target-site resistance arising is to target sequences that show high levels of functional constraint and can therefore not easily tolerate variant alleles[1]. We recently demonstrated the success of this approach by developing a second generation gene drive, herein named Ag (QFS)1 (previously called *dsxF*^*CRISPRh*), that has been used to suppress entire populations of caged mosquitoes in proof-of-principle experiments[9]. This gene drive is designed to target an ultra-conserved, essential sequence within the female-specific isoform of the gene *doublesex*, encoding a transcription factor that is the major regulator of sex determination in insects[9,10]. Females homozygous for the gene drive display female-male sexual development (intersex) and cannot produce offspring. This strategy has proven effective for two independent gene drive designs, each tested by tracking invasion dynamics over time following single, low frequency introductions in six discrete-generation laboratory populations[9,11].

Typically, the development of candidate gene drive strains for potential vector control involves assessment of basic parameters concerning both fitness and drive, such as the homing rate, life-span and fecundity, however these parameters are notoriously difficult to estimate and often context-specific. Promising strains are then tested to determine if the gene drive can spread in small caged populations, and to compare invasion dynamics with prediction. This initial testing is key for identifying promising candidate gene drive strains, however it provides information of limited predictive value as these experiments do not take into account age-structured populations, complex mating behaviours, differing probabilities of finding food resources, oviposition sites, and mating opportunities. Indeed, previously developed genetically modified mosquito strains have shown strong fitness costs when tested in large-cage or semi-field experiments that were not observed in initial small cage testing[12], including severe mating disadvantages that precluded further testing of the strain[13]. We refer herein to the initiation of these indoor large-cage experiments with the gene drive strain as releases, given that this is what they are designed to emulate, albeit they are performed in fully contained chambers that comply with appropriate arthropod containment guidelines. Many of these fitness challenges and complex behaviours can be reproduced in large cages[14] by allowing overlapping generations so as to reveal potential differences in life-span and fecundity over time that cannot be captured in discrete-generation studies[8,15,16]. As such, large-cage release experiments are now considered an essential bridge between laboratory and field testing within the tiered testing approach[4,15,17–19].

The Ag(QFS)1 strain is designed to make homozygous gene drive females infertile, and so it is dependent upon high fitness in males and in heterozygous carrier females (where the gene drive is designed to be active in the germline) to ensure it increases in frequency in the population. Initial testing of the strain revealed a reduction in the fertility of heterozygous females that is likely due to leaky activity of the gene drive in the soma, leading to a mosaic pattern of knockout of the *doublesex* target gene[9]. As *doublesex* plays a crucial role in the physiological development of females, this mosaicism may impact upon complex behaviours that are difficult or impossible to reproduce in small cages, including swarming, food and oviposition site searching, and resting.

Here, we present the results of four large-caged release experiments designed to challenge the suppressing activity of Ag (QFS)1 in an environment that partially mimics natural conditions and can invoke complex behaviours. We use an overlapping generation study design that is more likely to reveal differences in general fitness, mating success, and fecundity over time that cannot be captured in discrete-generation studies. Ag(QFS)1 males are introduced at ~12.5 or 25% initial drive frequency and key measurements of drive invasion and population fitness are monitored over time. We observe increases in frequency of the transgenic mosquitoes within the populations in all four cages initiated with the drive that lead to complete population suppression by 245–311 days after introduction. We compare these results to the output of a stochastic model using the method of Approximate Bayesian Computation, in order to infer key life-history parameters that are difficult to measure in dedicated assays. Our findings represent the first successful demonstration of efficacy for a gene drive in the second phase of testing which focuses on acquiring information under challenging ecological conditions, provide a platform for generating key evidence to inform initial go/no-go operational decisions, and pave the way for the first field trials of gene drive technology.

## Results

**Ag(QFS)1 spreads rapidly through age-structured mosquito populations in large cages.** After stabilising the receiving wild-type age-structured populations in the large cages (Fig. 1), we seeded the age-structured large cage (ASL) populations in duplicate with gene drive mosquitoes at 12.5 and 25% allelic frequencies of the estimated pre-released adult population size in the large cages. We also kept two ASL populations unseeded as controls. We were able to track the inheritance of the gene drive allele by virtue of the dominant RFP marker gene. We observed substantial variability in the rise in frequency of gene drive-positive mosquitoes, regardless of starting gene-drive frequency (Fig. 2g, h, Source Data). We observed apparent two-weekly fluctuations in the gene-drive frequency time-series, which were confirmed by analysis of temporal autocovariance; all four time-series exhibited positive autocovariance at a time lag of 2 weeks and negative covariance at shorter time-lags (Supplementary Fig. 4). We note that the fluctuations were not apparent in the model simulated time-series (Supplementary Fig. 4). The spread of the Ag(QFS)1 followed a sigmoidal pattern of invasion, increasing in frequency slowly for the first 100–150 days, followed by a rapid period of invasion, and finally slowing as the drive approached fixation between 220 and 276 days after introduction in the low frequency release cages (Fig. 2g) and between 224 and 241 days after introduction in the medium gene-drive frequency release cages (Fig. 2h). No gene-drive positive individuals were detected in control cages, consistent with the cages being fully isolated from one another (Supplementary Data 1).

**Increase in frequency of the gene drive allele causes suppression of ASL mosquito populations.** As Ag(QFS)1 approached fixation there was a rapid decline in the fraction of fertile females

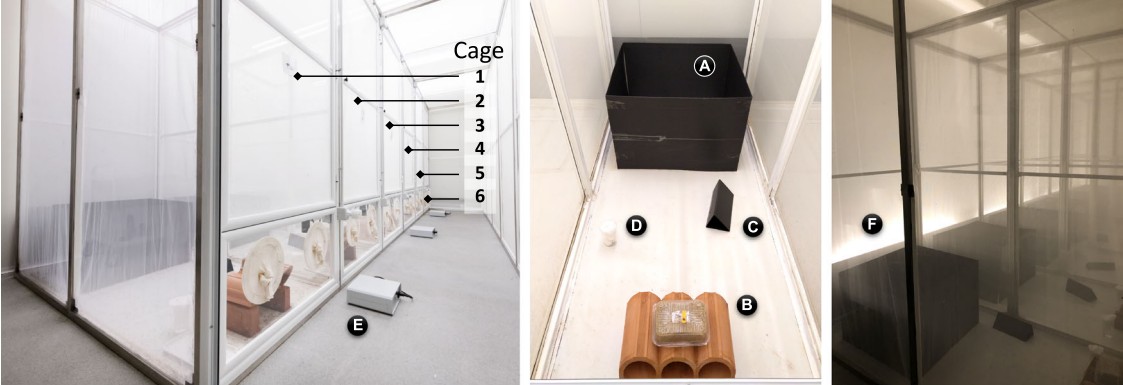

(A) Swarming arena. (B) Wet resting site (bricks). (C) Dry resting site. (D) Glucose feeder. (E) Hemotek blood feeding system. (F) Sunset simulation

**Fig. 1 Design of the large cages used in this study.** Images of the six large cages (numbered) within the climatic chamber (left panel) with the typical arrangement (central panel) of the swarming arena (**A**), wet (**B**) and dry (**C**) resting sites and sugar source (**D**). The age-tructured populations in six large cages served as control (cage 1 and 4) or were seeded with low frequency of Ag(QFS)1 (cages 2 and 5) and medium frequency of Ag(QFS)1 (cages 3 and 6). Also shown is the Hemotek feeding system (**E**) and the black horizon marker to emulate sunset (**F**, panel on the right). For blood feeding, two Hemotek feeders were introduced in each cage through one of the two openings at the front, leaving the power unit outside. Source: No Source Data.

as the growing proportion of gene drive homozygotes, lacking a functional copy of the female isoform of the *doublesex* gene, develop into sterile 'intersex' adults (Fig. 2d, e). As the formation of homozygotes is a requirement for population suppression, a strong and unambiguous reduction in egg output occurred only after the frequency of the gene drive allele rose above 90%, culminating in complete population suppression 245–311 days after release of Ag(QFS)1 in the low gene-drive frequency release cages (Fig. 2a) and by days 266–276 in the medium gene-drive frequency release cages (Fig. 2b). By comparison, the mosquito populations in the control cages maintained a stable sex ratio (Fig. 2f) and an average of more than 10,000 eggs over the final month of the experiment (Fig. 2c), while ASL populations seeded with Ag(QFS)1 collapsed.

**Similar adult longevity of Ag(QFS)1 and wild-type strains.** No significant differences in adult survival between of Ag(QFS)1 and wild-type strains were detected in large cages ($P = 1.0$, Kruskal–Wallis test), with 50% median mortality at day 6 (95% CI = 5–6 days) and day 11 (95% CI wild-type = 9–13 days, 95% CI Ag(QSF)1 = 11–12 days) at the beginning and the end of the large-caged release experiment, respectively (Supplementary Fig. 3 and Source Data). No difference in male and female survival were observed for G3 in both small and large cages and for Ag(QFS)1 in small cages. A small difference was observed between Ag(QFS)1 homozygous and heterozygous individuals in both small and large cages before the population experiment. Overall, survival in large cages is substantially lower than in small cages maintained under similar environmental and rearing conditions, where 50% mean mortality occurred at 20 days. In agreement with Pollegioni et al.[16] our data suggest that females survive longer than males when housed in large cages.

We observed an increased adult longevity in the large cages after the year-long experiment compared to before the Ag(QFS)1 release (median of 11 days and 6 days, respectively; $P = 0.032$, Kruskal–Wallis test) irrespective of the genotype. Individuals reared in the small cages tested in the same conditions (after the year-long experiment) showed the same adult survival than those collected from the ASL populations (for both G3 wild type and Ag(QFS)1 transgenics), suggesting the difference is due to the micro-environmental conditions of the large cages and not due to strain adaptation or the genotypes.

**Parameter inference reveals drive allele female fertility costs in age-structured mosquito populations.** The posterior mean density for the fertility of females whose father was transgenic was 0.35 (indicating a 65% reduction in egg output relative to wild-type females), with a 95% credible interval of (0.17–0.57) (Fig. 2i; Supplementary Table 1). By contrast, the marginal posterior distribution for the fertility of females whose mother was transgenic closely resembled its prior distribution, indicating a lack of statistical power to infer this parameter from the ASL data. This reflects the relative rarity of such females, due to both homozygous female sterility and also the heterozygous fitness costs themselves, whereby the cage dynamics are insensitive to their fertility. It is therefore not possible to determine whether or not both types of female offspring differ in fertility on the basis of this data, and it is also not possible to discern the relative roles of parental effects and Cas9 deposition on female fertility. However, the pairwise posterior probabilities of the two parental parameters have negative covariance, indicating that additional information on one parental effect would enable the other to be more accurately determined (Supplementary Fig. 5).

The posterior densities indicated that females typically lay around 116 eggs per batch (51–213), and around 14% of mated females laid eggs at each twice-weekly opportunity (8–21%). The posterior mean density for the fraction of non-homed gametes produced by heterozygous individuals becoming non-functional resistance alleles was around one half (50%; 27–83%).

**Stochastic simulations capture dynamics of spread and suppression.** Simulations of the cage dynamics using parameters drawn at random from the posterior distribution closely corresponded to the observed trends in the frequency of drive-carrying individuals (Supplementary Fig. 6g, h). This is expected, since the posterior distribution was inferred from the data, yet it gives confidence that the model captures much of the biology of the cage population. The simulations performed less well in replicating the variability in egg laying in the control cages, suggesting the model does not incorporate all the sources of this variation (Supplementary Fig. 6c). We ran 1000 simulations of the posterior informed model to predict the range of potential cage dynamics. All simulations ended with complete population suppression within 560 days, and 95% of the simulations reached this state within 399 or 329 days for the low and high gene-drive frequency releases, respectively (Supplementary Fig. 7).

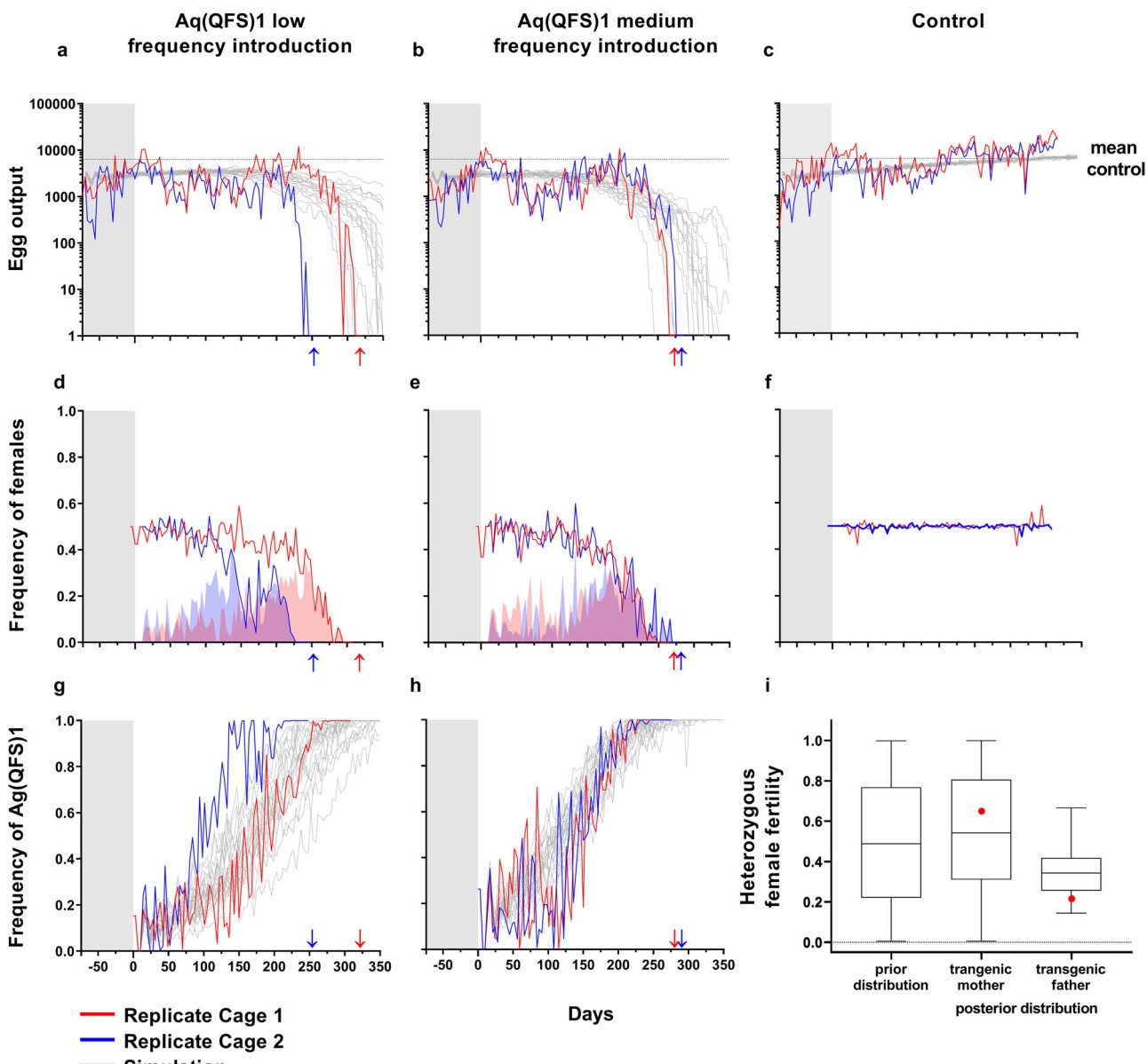

**Fig. 2 Kinetics of spread of Ag(QFS)1 in age-structured large cage populations.** We clarified the legend as following: Age-structured large cage (ASL) populations were established over a period of 74 days (shaded grey) and seeded in duplicate with Ag(QFS)1 heterozygous males at low (12.5%, **a**, **d**, **g**) and medium (25%, **b**, **e**, **h**) allelic frequency, whereas two control ASL populations were maintained without introduction of the Ag(QFS)1 gene drive (**c**, **f**). The total egg output (**a**, **b**, **c**), the total frequency of females with apparently normal external morphology (i.e. wild type and heterozygous) (**d**, **e**, **f**), and the frequency of Ag(QFS)1 alleles (**g**, **h**) were monitored over time (red and blue lines for replicate cages). Mean egg output of the control is indicated by a dashed line (**a**, **b**, **c**). Red and blue shaded areas indicate the fraction of morphological females that carried the gene drive in heterozygosity (red), or were wild type (blue) (**d**, **e**). Arrows indicate the point at which no further eggs were recovered, the point at which populations were considered eliminated. A total of 20 stochastic simulations of the egg output and the frequency of Ag(QFS)1 (grey lines) were modelled using default parameters based on Kyrou et al.[9] and expert judgement (Supp. Methods), superimposed to experimental data for the control and gene drive introductions (**a**, **b**, **c**, **h**). $n = 200$ samples from the prior and posterior distribution of the relative fertility of Ag(QFS)1 heterozygous females that putatively received deposited nuclease paternally or maternally, as compared to the average fertility of wild-type females (**i**). Fertility distributions are represented as boxplots where the centre line denotes the median relative fertility (50th percentile), lower and upper bounds of the box contain the first and third quartiles, and whiskers mark the minimum and maximum values. Shown in red are the estimates of female fertility from experimental observation in Kyrou et al.[9]. Source: Dataset 1.

**Drive-resistant alleles were not generated in Ag(QFS)1 seeded age-structured mosquito populations.** To investigate whether drive-resistant alleles had been generated or selected as the gene drive allele increases in frequency in the ASL populations, we performed pooled amplicon sequencing around the gRNA target site on samples of the larval progeny (150–1200/cage) collected at early and late timepoints after release (Fig. 3). These alleles can take two forms: functional resistant alleles that restore

a viable gene product, and non-functional resistant alleles that do not. Resistant alleles may be pre-existing in the population or generated by the gene drive itself as a result of error-prone end-joining. In spite of the incredible selective pressure exerted by Ag (QFS)1, no mutant alleles were generated that could conceivably code for a functional DSX protein.

We identified three putative end-joining mutations present above the sequencing threshold frequency[20] of 0.25% in any of

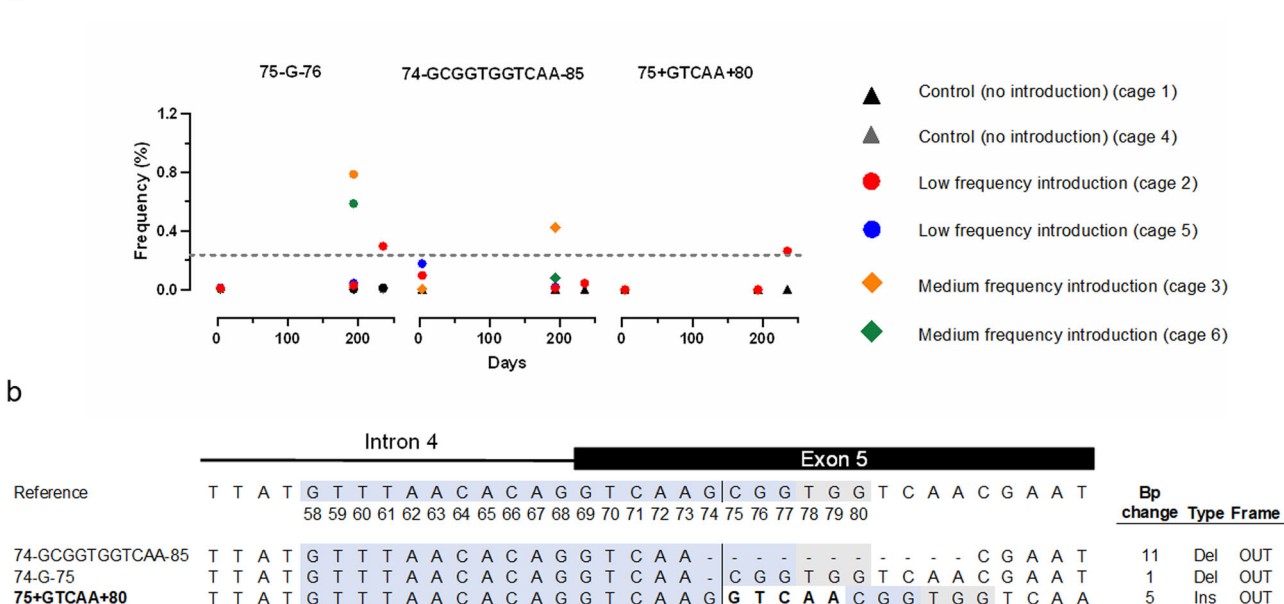

**Fig. 3 Drive-resistant mutations do not come under positive selection as Ag(QFS)1 spreads in age-structured mosquito populations. a** The % frequency of three putative non-restorative resistant alleles (R2) (75-G-76, 74-GCGGTGGTCAA-85, and 75+GTCAA+80) detected above the threshold frequency of 0.25% (Pfeiffer et al.[20]), in at least one cage at a single point in time, amongst all non-drive alleles, is shown over time. Samples were taken on days 4 and 193 for all cages, and on day 235 where the number of mosquitoes exceeded the re-stocking requirement. The naming of each mutation indicates the base pairs inserted (+) or deleted (−), and its location relative to the Cas9/gRNA cleavage site between position 74 and 75, depicted in (**b**). Low gene-drive frequency introduction cages 2 and 5 were initiated at a maximum Ag(QFS)1 allelic frequency of 12.5%, whilst medium frequency introduction cages 3 and 6 were initiated at 25%. Wild-type control cages 1 and 4 did not contain Ag(QFS)1. **b** The position of the three R2 alleles detected is shown, and compared to the reference *An. coluzzii* and *An. gambiae* sequence of the intron4/exon 5 junction of the *doublesex* gene. Highlighted nucleotides indicate the gRNA binding site (blue) and PAM sequence (grey). Inserted nucleotides are shown in bold. The number of base pairs inserted or deleted and the effect on the resulting allele (in-frame (IN), or out-of-frame (OUT)) is shown to the right. Source: Dataset 1.

the four release cages. All three alleles introduce a frameshift mutation that would disrupt the female isoform of *doublesex*, including a 5-bp insertion that was uniquely identified in this study and two deletions (1 bp and 11 bp in length) that were previously identified in small caged testing of Ag(QFS)1[9]. The failure of any of these alleles to spread above 1% frequency amongst non-drive alleles would suggest they are highly deleterious and undergo no positive selection as the gene drive allele increases in frequency.

## Discussion

In this study we provide evidence that the *doublesex*-targeting gene drive strain, Ag(QFS)1, is able to effectively suppress age-structured populations reared in an environment that recapitulates some parameters typical of natural conditions and induces some mosquito behaviours observed in the field. This gene drive has previously been demonstrated to spread effectively through populations of wild-type *An. gambiae* mosquitoes maintained in small cages (0.0156 m³) with non-overlapping generations[9]. We observed similar dynamics of spread in duplicate age-structured populations in large cages (4.7 m³) cages initiated with low or medium frequency of the drive, leading to complete population suppression within 245–311 days. In addition to the overarching dynamics of spread, we observed marked fluctuations in drive frequency in all four experimental cages where releases took place. These fluctuations suggest that interbreeding between young and old cohorts of cohabiting adults may be rare under large-cage conditions, though further investigation would be needed to confirm this hypothesis. The fluctuations were not apparent in the model simulated time-series, yet we found that both stochasticity and dynamics of spread were largely explained

by modelling predictions based upon comprehensive characterisation of the life-history traits Ag(QFS)1 (Fig. 2).

Retrospective inference of life-history parameters from cage population data allows a deeper insight into the phenotypic effects of transgenes, beyond what one can learn from small cage studies alone[16,21]. This analysis suggests that female fertility is the most important parameter that determines the dynamics of this gene drive. The simulations based on previous small cage data[9] alone (Fig. 2) corresponded to the observations almost as well as the retrospective informed simulations (Supplementary Fig. 6), probably because the single generation measurements of female fertility gave similar results to the inference from the large-cage data. This in itself suggests that the costs to female fertility conferred, at least by this gene drive targeting the female isoform of *doublesex*, may be quite stable within the environmental conditions in which mosquitoes are reared. Moreover, the accuracy of the prior simulations indicate that this drive allele confers few, if any, fitness effects in the semi-field environment that were overlooked by the small cage studies (with the exception of adult survival). Whether this holds for future gene drive designs, and which aspect of the resulting phenotype conferred has the largest effect on the veracity of predicting its trajectory in a population, will depend on the nature of the gene drive element and its molecular target.

A previous study found that the fitness of drive-heterozygous females was dependent on which parent contributed the drive allele[9], and two explanations were given. The cost may be due to paternal and maternal effects of Cas9 deposition into the sperm or egg, or it may result from ectopic activity of Cas9 in the soma, rather than the germline. Both possibilities, which are not mutually exclusive, will lead to suboptimal fitness due to a mosaic

pattern of disruption of the *doublesex* gene. However, they may have subtly different ramifications to the potential spread of the drive allele in natural populations, since parental deposition will affect all offspring of heterozygous parents while ectopic activity will only affect offspring with the gene drive[22,23]. We were unable to identify the potential causes of fertility cost from our analysis of the large-cage observations, which reflects the relatively modest differences in their effects. Such disentanglement is perhaps easier to achieve from small cage studies where specific genotypes are crossed. By this method, Kyrou et al.[9] found that females had lower fitness if descended from a transgenic father than transgenic mother, suggesting a potential role of paternal Cas9 deposition, a phenomenon that we have previously observed in other transgenic strains expressing a nuclease during spermatogenesis[2,24]. Additionally we cannot exclude other biological mechanisms that may be related to the position of the gene drive nuclease within a sex determination gene.

As with other forms of vector control, gene drives designed for population suppression will exert a strong selection for resistance[6]. The force of selection for resistant mutations is proportional to the fitness cost imposed by the gene drive itself but it can apply even to population modification gene drives that are intended to drive an anti-parasitic effector gene into a vector population, with the intention of changing its competency to transmit pathogens[25]. The most likely form of resistance is a change in the target sequence that can prevent cleavage by the nuclease. Various strategies exist for reducing the probability of resistance arising against both population suppression and population modification gene drives. In the case of Ag(QFS)1 the gene drive is deliberately designed to target a region of its *doublesex* target gene that is under high functional constraint and cannot readily generate or accommodate sequence variants that confer functional resistance.

No functional resistant alleles evolved in our previous small cage experimental studies of gene-drives targeting *doublesex*, demonstrating the promise of this strategy[9,11]. The large-cage experiments presented here probably provided even greater selective pressure for resistance, both because of their longer duration (245–311 days after initial release), and also their potential to reveal additional fitness costs such as complex mating and oviposition behaviours. In spite of this pressure and a concerted effort to identify resistant alleles, none were found to be capable of restoring the function of *doublesex*.

Indeed, we identified just three mutant alleles that were each unable to encode a functional DSX protein and present at low frequency (<1% amongst non-drive alleles). Somewhat surprisingly, fewer non-functional mutant alleles were detected in our large semi-field cages than in the previous small caged release experiments (Fig. 3). This may be due to the harsher environment of the large cages that results in a stronger purifying selection against non-functional resistant alleles, or it may simply reflect differences in the effective population size, which have a similar effect in reducing the variety of available alleles. Though these non-functional resistant alleles cannot completely displace a gene drive, modelling suggests that under specific permissive conditions they can compete to reach a stable equilibrium (that nonetheless results in a strong and sustained population suppression) (Beaghton et al.[22]), an outcome we found neither in caged releases of Ag(QFS)1 nor in 1000 stochastic simulations (Supplementary Fig. 7). Large population sizes and low release rates increase the probability of these equilibriums forming; conversely, high frequency releases and multiplexed/combined drives can mitigate against it. Further studies must specifically address the probability of resistance, either naturally occurring or generated by the nuclease, to predict the potential spread, suppression and operational lifetime of Ag(QFS)1.

This study is the first successful test of gene drive technology in age-structured populations in an environment that mimics natural conditions and can invoke complex behaviours, and thus represents an essential intermediate step to move gene drive technology from laboratory studies to the field. Our data generated in the more realistic ecological setting in large cages, allowing the mosquitoes to show a complex feeding and reproductive behaviour, can inform go/no-go decisions by reducing uncertainty on the efficiency of gene-drive modified mosquitoes.

In accordance with the Code of Ethics for Gene Drive Research[26], we have established a paradigm for generating data that helps to bridge lab and field studies. Given their transformative potential, proposed pathways to the deployment of gene drive mosquitoes have been the subject of much discussion recently, yet all recommend a staged, step-by-step pathway, that moves through various levels of confinement prior to testing in an open release setting[4,19,26,27]. For a gene drive designed to be highly invasive and with a very low threshold of invasion, such as the one described here, its testing in large indoor cages with overlapping generations, designed to mimic more closely the native ecological conditions is fundamental to proving its efficacy, in a safe manner[4,19]. In the future, further improvements to the cage design could be made, such as establishing more realistic conditions for aquatic life stages that more faithfully recapitulate larval competition. Nonetheless, the Ag(QFS)1 strain is the first gene drive strain to pass this essential intermediate step within a tiered testing approach and, whilst comprehensive resistance testing and environmental risk assessment will be needed ahead of field trials[17,28], confirms that gene-drive modified mosquitoes show great promise as a tool for vector control.

## Methods

**Study design.** Initially, we assessed life-history traits of both Ag(QFS1) males and females as well as of the wild-type strain G3 of *An. gambiae* and assessed their longevity under large-cage conditions (4.7 m³) in order to emulate more natural population dynamics[16] (see Fig. 2, Supplementary Material). Considering the initial Kaplan–Meier Survival estimate of wild-type G3 adult mosquitoes in 4.7 m³ cages of 2 m × 1 m × 2.35 m size and the establishment of overlapping generations with bi-weekly introductions of 400 G3 pupae with a start-up population of 800 mosquitoes, we then analysed ASL populations with an expected mean size of ~570 adult mosquitoes as 'receiving' populations for gene drive release experiments (Source Data). To mimic field-like conditions absent in small cage conditions, the climate chambers were maintained under near-natural environmental conditions including simulated dusk, dawn and daylight, and each cage was equipped with proven swarming stimuli and a resting shelter[14] (Fig. 1). Under these conditions male swarming, an important component of successful mating behaviour, was frequently observed. To mimic a hypothetical field gene drive release, we seeded Ag(QFS)1 mosquitoes over a single week (two releases) into the established 'receiving' wild-type populations at two different starting frequencies, low (12.5% initial allele frequency) and medium (25% allele frequency), as well as control cages (0% gene drive release), all in duplicate (6 cages total). The ASL population dynamics and the potential selection of drive-resistant alleles were monitored in treated and control cages until wild-type populations were fully suppressed by the gene drive in the treatments. Finally, we constructed an individual-based stochastic simulation model of the experiment to better understand the observed dynamics of the gene drive frequency and population suppression.

**Mosquito strains.** Two *An. gambiae* mosquito strains were used, the wild-type G3 strain (MRA-112) and Female Sterile Gene Drive strain, Ag(QFS)1, previously known as dsxF^CRISPRh[9].This strain contains a Cas9-based homing cassette within the coding sequence of the female-specific exon 5 of the *dsx* gene (Supplementary Fig. 1). The cassette includes a human codon-optimised *Streptococcus pyogenes Cas9 (hSpCas9)*[29] gene under the regulation of the *zero population growth (zpg)* promoter and terminator[30] of *An. gambiae* and a gRNA against exon 5 under the control of the *An. gambiae* U6 snRNA promoter. The cassette also carries a dsRed fluorescent protein marker under the expression of the 3xP3 promoter.

**Mosquito containment and maintenance.** *Anopheles gambiae* mosquito strains were contained in a purpose-built Arthropod Containment Level 2 plus facility at Polo d'Innovazione di Genomica, Genetica e Biologia, Genetics & Ecology Research Centre, Terni, Italy. Mosquitoes were reared in cubical cages of 17.5 cm × 17.5 cm × 17.5 cm (BugDorm-4) as described in Valerio et al.[31] at 28 °C and 80% relative

humidity (Supplementary Fig. 2). Larvae were maintained in trays (253 × 353 × 81 mm) at a density of 200 larvae per tray using 400 mL deionized water with sea salt at a concentration of 0.3 g/L and 5 mL of 2% w/v larval diet[32] and screened for fluorescent markers en masse using a Complex Object Parametric Analyzer and Sorter (COPAS, Union Biometrica, Boston, USA).

**Large-cage environment**. For experimental purposes, mosquitoes were housed in a large-cage environment as described in Pollegioni et al.[16] A single large climatic chamber was equipped with six 4.7 m³ cages of 2 m × 1 m × 2.35 m (length, width and height) (Fig. 1) and maintained at 28 °C ± 0.5 °C and 80 ± 5% relative humidity (Fig. 1, Supplementary Fig. 2). The climatic chamber was illuminated by three sets of three LEDs (3000, 4000 and 6500 K correlated colour temperatures) controlled by Winkratos software (ANGELANTONI Industries S.p.A, Massa Martana, Italy), allowing a gentle transition between light and dark sufficient to emulate dawn, and dusk. For the purpose of the current study, full light conditions (800 lux) were simulated using all LEDs and adjusted to last 11 h and 15 min. Cages were additionally equipped with ambient lighting (3000 K) designed to stimulate swarming[14], and a terracotta resting shelter moistened with a soaked sponge. Mosquitoes were fed on 10% sucrose and 0.1% methylparaben solution and blood fed bi-weekly using defibrinated and heparinized sterile cow blood via the Hemotek membrane feeder (Discovery Workshops, Accrington, 34 UK). Oviposition sites consisted of a 12 cm diameter Petri dish with a wet filter paper strip introduced 2 days after the blood meal. Mosquito pupae, food, blood and water were introduced or removed through two openings, 12 cm in diameter, at the front of each cage with no operators entering the cage. Blood meal was administered by the introduction of two Hemotek feeders in each cage through one of the two openings at the front, leaving the power unit outside. No adult mosquitoes were removed from the large cages throughout the cage trials.

**Measuring the life-history parameters**. To assess life-history parameters of wild-type G3 and Ag(QFS)1 strains, standardised phenotypic assays were performed as described in Pollegioni et al.[16] In brief, clutch size, hatching rate, larval, pupal and adult mortality rates, as well as the bias in transgenics among the offspring of heterozygous Ag(QFS)1 were measured in wild-type G3 and Ag(QFS)1 strains in triplicate in standard small laboratory cages (BugDorm-4). Ag(QFS)1 heterozygotes used in these assays had inherited the drive allele paternally and were therefore subject to paternal, but not maternal, effects of embryonic nuclease deposition that can lead to a mosaicism of somatic mutations at the doublesex locus and a resultant effect on fitness[9]. 150 females and 150 males were mated to wild-type mosquitoes for 4 days, blood fed and their progeny counted as eggs using EggCounter v1.0 software[33]. Hatching rate was evaluated 3 days post oviposition by visually inspecting 200 eggs under a stereomicroscope (Stereo Microscope M60, Leica Microsystems, Germany). Sex-specific larval mortality was calculated by rearing 200 larvae/tray and counting/sexing the number of surviving pupae.

Sex-specific adult survival was assessed in triplicate for each genotype separately by introducing and sexing 100 male and 100 female pupae of G3 and heterozygous Ag(QFS)1 into either small (0.0049 m³) or large cages (4.7 m³) (Supplementary Fig. 3). In the small cages, we tested 100 individuals in each cage divided by genotype and sex. In each large cage, 100 male and 100 female pupae following sexing and counting were tested together. Because homozygous Ag(QFS)1 do not show clear sex-specific phenotypes as pupae[9], 100 Ag(QFS)1 total homozygotes (males and intersex females) were introduced into the small and large cages unsexed (Supplementary Fig. 3a). Sex-specific survival of emerged adults was calculated from daily collections of dead adult mosquitoes from the respective cages and their sexing. The adult survival assays in large cages were performed twice, one before the large-cage Ag(QFS)1 release experiment started and one after the large-cage Ag(QFS)1 release experiment finished. For the latter adult survival assay, around 400 individual mosquitoes were collected from large-cage populations at larval stage (before the cage populations declined, day 231 and 311 post-release for Ag(QFS)1 and G3 wild type, respectively), and kept in small cages until the start of the assay (Supplementary Fig. 3b).

**Establishment, maintenance and monitoring of age-structured large cage (ASL) populations**. To test the suppressive potential of Ag(QFS)1, we first established stable ASL populations of An. gambiae (G3 strain) housed in a purpose-built climatic chamber. Each population was initiated and maintained at the maximum rearing capacity through twice-weekly introductions of 400 G3 pupae (200 males and 200 females) over a period of 21 days (establishment), estimated to sustain a mean adult population of 574 mosquitoes based on the initial Kaplan–Meier estimate (Supplementary Fig. 3a). After this initial period only progeny of these populations were used to repopulate the cages twice-weekly (re-stocking) for a period of 53 days (pre-release, 74 days total), or supplemented with wild type reared separately when progeny numbers were too low. Each ASL population was considered stabilised after retrieving a sufficiently large and stable number of eggs to restock the population over four consecutive weeks. In detail, the receiving populations in all six cages were stabilised to produce a similar number of eggs in the 31 days before Ag(QFS)1 release, with an average egg production per cage ranging from 2262 to 5334. Twice-weekly blood meals were initiated at dusk and extended for a period of 5 h, and oviposition sites were illuminated with blue

light for egg collection 2 days later. Eggs were removed from the cages, counted, and allowed to hatch in a single tray within the climatic test chamber. For re-stocking the cage populations with wild-type pupae, a maximum of 400 randomly selected pupae were collected at the peak of pupation, manually sexed and screened and introduced to their respective cage twice per week.

**Ag(QFS)1 release experiments in large cages**. To assess invasion dynamics of the Ag(QFS)1 strain in ASL populations of Anopheles gambiae, we performed duplicate releases designed to randomly seed ASL populations at low (12.5%, cages 2 and 5) or medium (25%, cages 3 and 6) allelic frequencies. After 74 days pre-release initiation period, heterozygous Ag(QFS)1 males were released into duplicate cages in addition to the regular re-stocking of the ASL populations with wild-type pupae. Releases took place on two consecutive re-stocking occasions, representing 15.2% (71 and 72) or 26.3% (142 and 143) of pupae introduced that week (943 and 1085, respectively), equivalent to 25 or 50% of the estimated mean pre-released adult population (on average 574 mosquitoes were present in large cages). No further releases were carried out and indoor ASL populations were maintained through re-stocking of 400 pupae twice per week. From then, the ASL populations were maintained in the same way we established the receiving population, with the same constant re-stocking rate from offspring. No adult mosquitoes were removed from the cages. Duplicate control cages were similarly maintained, but without release of Ag(QFS)1.

While not statistically significant (Kruskal–Wallis Test $P = 0.06$ ns), there was some variation in reproductive output amongst the six cages due to random effects (cage 1: mean egg number = 4265.77, CI 95% = 1550.36; cage 2: mean egg number = 2691.73, CI 95% = 790.41; cage 3: mean egg number = 2517.46, CI 95% = 889.66; cage 4: mean egg number = 1799.18, CI 95% = 573.18; cage 5: mean egg number = 2350.82, CI 95% = 745.44; cage 6: mean egg number = 2060.05, CI 95% = 767.77). To control for random effects that could affect reproductive capacity of the population independently of the effect of the gene drive, we chose as control populations those cages with reproductive output at the upper and lower end of the distribution (cages 1 and 4). Replicate gene-drive release cages were distributed to cages 2 and 5 (12.5% allelic frequency) and cages 3 and 6 (25% allelic frequency) to mitigate against potential local environmental position effects (Fig. 2).

Key indicators of population fitness and drive invasion were monitored for the duration of the experiment, including total egg output, hatching rate, pupal mortality, and the frequency of transgenics amongst L1 offspring and the pupal cohorts used for re-stocking. Total larvae were counted and screened for RFP fluorescence linked to Ag(QFS)1 using the COPAS larval sorter, and 1000 pupae selected to rear at a density of 200 per tray. Pupae positive for the gene drive element could be identified by expression of the RFP marker gene that is contained within the genetic element. Triplicate samples of up to 400 L1 larvae were stored in absolute ethanol at −80 °C for subsequent analysis.

**Modelling**. A stochastic model was set up to replicate the experimental design with respect to twice-weekly egg laying, the initiation phase, the transgene introductions, and the subsequent monitoring phase (Supplementary Methods). In brief, daily changes to the population result from egg laying, deaths, and matings, and are assumed to occur with probabilities that may be genotype specific. Adult longevity parameters were estimated from the large-cage survival assays that were performed before the gene-drive release experiments began, and after the gene-drive dynamics had run their course. The ASL caged populations showed a similar trend of increasing egg output over time prior to the suppressive effect of the drive (Fig. 2a–c) that may be explained by a general increase in adult survival that was observed between the start and end of the population experiment (Supplementary Fig. 3). To account for these changes in the stochastic model, we assumed a small increase in adult survival over time, irrespective of genotype, based on experimental data (Supplementary Fig. 3).

We were particularly interested in the drive allele fertility costs, because these are potentially important to drive allele dynamics in natural populations[22,23]. Fertility costs may arise from paternal and maternal effects of Cas9 deposition into the sperm or egg, or from ectopic activity of Cas9 in the soma[9]. It is therefore possible that female offspring of transgenic fathers differ, in terms of fertility, from female offspring of transgenic mothers, and to investigate this possibility we fitted a separate parameter for the fertility of each type of female.

We compared the data to model simulations using a suite of summary statistics[34] (Supplementary Methods) to infer the fertility of females with a transgenic father or mother. In addition, we inferred two parameters that determined the egg production of unaffected (wild-type) females, and one parameter that determined the rate of R2 allele creation. We obtained a posterior distribution for all five parameters by retaining the 200 best fitting parameter combinations from 50,000 parameter samples generated by a Monte-Carlo algorithm (Supplementary Table 1). The simulation codes are available from Github: https://github.com/AceRNorth/TerniLargeCage.

**Pooled amplicon sequencing and analysis**. We previously developed a strategy to detect and quantify target-site resistance based upon targeted amplicon sequencing using pooled samples of larvae[6], and found no evidence for resistance to Ag(QFS)1 in small caged release populations[9]. To further investigate resistance in the large-caged

release experiment, we analysed mutations found at the genomic target of Ag(QFS)1 in samples collected at early and late timepoints. Genomic DNA (gDNA) was extracted *en masse* from triplicate samples of 400 L1 larvae, or 50–300 larvae where larval numbers were limiting, that were collected after blood meals given on days 4 and 193 from all 6 cages, and on day 235 where sufficient larvae were available.

gDNA extractions were performed using the DNeasy Blood & Tissue kit (Qiagen). 100 ng of extracted gDNA was used to amplify a 291 bp region spanning the target site of Ag(QFS)1 in *doublesex*, using the KAPA HiFi HotStart Ready Mix PCR kit (Kapa Biosystems) and primers containing Illumina Genewiz AmpEZ partial adaptors (underlined): Illumina-AmpEZ-4050-F1 ACACTCTTTCCCTACACGACGCTCTTC CGATCTACTTATCGGCATCAGTTGCG and Illumina-AmpEZ-4050-R1 GACT GGAGTTCAGACGTGTGCTCTTCCGATCTGTGAATTCCGTCAGCCAGC. PCR reactions were performed under non-saturating conditions and run for 25 cycles, as in Hammond et al.[6] to maintain proportional representation of alleles from the extracted gDNA in the PCR products.

Pooled amplicon sequencing reads, averaging ~1.5 million per condition, were analysed using CRISPResso2[35], using an average read quality threshold of 30. Insertions and deletions were included if they altered a window of 20 bp surrounding the cleavage site that was chosen on the basis of previously observed mutations at this locus[9]. Individual allele frequencies were calculated based upon their total frequency in triplicate samples. A threshold frequency of 0.25% per mutant allele was set to distinguish putative resistant alleles from sequencing error[20].

**Reporting summary**. Further information on research design is available in the Nature Research Reporting Summary linked to this article.

## Data availability
The data generated in this study are provided as Supplementary Figures and Tables and in the Supplementary Information. The raw data generated in this study are also available in the DRYAD database [https://doi.org/10.5061/dryad.9w0vt4bg0]. Source data are provided with this paper.

## Code availability
The mathematical algorithm that is deemed central to the conclusions is available in Supplementary Methods. The simulation codes are available from Github: https://github.com/AceRNorth/TerniLargeCage. Source data are provided with this paper.

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

## Acknowledgements
We would like to thank Austin Burt, Silke Fuchs, John Mumford and John B. Connolly for their valuable comments and suggestions to improve the quality of the paper. Supported by a grant from the Bill & Melinda Gates Foundation and from the Open Philanthropy Project Fund, an advised fund of Silicon Valley Community Foundation.

## Author contributions
Conceptualisation: A.H., P.P., T.N., R.Mu and A.C.; Methodology: A.H., P.P., T.P., A.N., A.S., R.Mu, T.N., and A.C.; Investigation: P.P., T.P., A.N., R.Mi, A.T., A.B., K.K., I.M., A.S. and R.Mu; Formal analysis A.H., P.P., T.P., I.M., A.N., A.S., T.N. and R.Mu; Project Administration: T.N. and R.Mu; Writing—original draft: A.H., P.P., A.S., A.N., T.N., R.Mu. Supervision: T.N., R.Mu and A.C.; Writing—review and editing: A.H., P.P., A.S., A.N., T.N., R.Mu, A.C.; Resources: A.C.; Funding acquisition: A.C.

## Competing interests
Authors have patents pertaining to the use of gene drives targeting DSX in arthropods (A.H., A.C. and K.K.) and using the zpg promoter for gene drive (A.H., A.C. and T.N.). Other authors declare no competing interests.
