## [Peer Review File · Nature Communications]

Reviewers' Comments:

Reviewer #1:

Remarks to the Author:

The work by Hammond et. al recapitulates earlier small cage trial experiments (non-overlapping discrete generations) in larger cage trials (continuous generations) mimicking conditions in the natural environment, and while they do show invasion/fixation/ and complete population suppression within a year (impressive). They also reveal a much more complicated ecology than was initially observed in earlier trials. Importantly, they did not detect drive-resistant alleles which would be expected as population suppression should exert a strong selection for resistance which would inhibit the spread of the drive. These are the first set of continuous generation gene drive experiments in an environment that mimics natural conditions bringing the technology one step closer to confined field-like conditions. Overall the experiments are well-designed, the manuscript is easy to read and I believe the results are of high importance. I don't have any major criticisms of this work - only have a few minor comments listed below.

Line by line Minor comments:

22: the term "self-sustaining genetic elements" is a very vague term and not necessarily the most precise way to describe GDs in the first sentence.

Line 22-24: This sentence is very long, bordering on run on. There are many different ideas tied together here. Consider rephrasing.

29: suppressing vs suppression?

49: needs a reference

52: If others have adopted progressively rigorous testing protocols, you should cite.

59: Concept of resistant alleles needs a reference.

79: "Since this is what they are designed to emulate" use of 'since' seems non-scientific, consider rephrasing.

77-82. Run-on. Rephrase.

Line 116: it is not clear how referencing a supplementary figure (S.Fig 1) with a the plasmid map is relevant here when the sentence implies life history traits would be outlined in this figure according to the text.

Sup Fig 1: Does the transgene have a terminator for Cas9? If so it should be annotated.

Sup Fig 1: Introns annotated with the standard 'pointing up carrot style' would be appreciated in the figure for clarity.

Sup Fig 1: CDS annotated... this is outside the CRISPR cassette, so is this the remainder of Exon 5? If so, a clearer label would be appreciated. If not, then this should be made clearer in the figure as well as legend.

Line 116: A reference for the Kaplan-Meier Survival quantities would be appreciated. Or if not available, a clearer description of what exactly was done here.

125: How was swarming induced? The prior sentence stated the swarming stimuli was in the cage already. Was it removed and replaced on a regular basis to 'induce' swarming? Or was swarming merely consistently observed in the cages due to permanent inclusion of the swarming stimuli? Clarification here would be appreciated.

Figure 1: The Hemotek Feeding system is shown outside of the cage. How was this configured when feeding occurred? Was the whole system inserted? Or were the membranes merely held up to the side of the cage? Or other? Clarification here would be appreciated.

Sup Fig 1: A 2nd chromosome map showing the wt dsx gene map, and gRNA target site would be appreciated for clarity.

Sup Fig 2: Inclusion of the temperature and humidity fluctuations were appreciated. I'm assuming the drastically decreased fluctuations observed between days ~140-230 were due to reduced staff entering and exiting the facility due to quarantine? If so, noting this somewhere in the legend or text would be appreciated, as this event represented a significant change to experimental conditions, and therefore should be noted. If not, and this reduction in ΔT or ΔH was due to some other change in the facility or operating error, it should also be noted.

Line 184: you mention that exactly 100 male and female were released into the caged, but then mention they were unsexed? To get these precise numbers, they must have been sexed at some point? Do you simply mean that both males and females (following sexing and counting) were released into the cage simultaneously? If so, please clarify.

Line 186: The AgQFS line does not show sexual dimorphism in pupae, but you released only 100 into cages? To be consistent with the cages described earlier which state 100 male and 100 female were used, should 200 AgQFS have been released? If not, please explain. If other, then this section could use some more explicit clarification.

Figure 2 line 9: "Putatively fertile non-intersex females (ie wt and het)" does this refer to the lines themselves, or the shaded areas? Please specify. Similarly line 11-12 "Red and blue shaded areas indicate fraction of non-intersex females carrying the gene drive in heterozygosity"... these two phrases state basically the same thing "non-intersex females" but refer to very different things on the graph (lines vs shaded areas). Is this a typo? Is one of these supposed to plot the frequency of wt? or true homozygous intersex? In particular the 'shaded' areas of the graph increase drastically just before the population crashes, so this shouldn't probably be referring to non-intersex, but instead true intersex? If not, then please clarify this legend.

Line 321: irrespective (not irrespectively). In current usage, it is an adjective describing the genotype

Line 346: "Means" is unscientific language. Suggest rephrasing. Perhaps something like "The overlap in the parent-specific estimates therefore precludes determination of parental sex effects on the fertility of transgenic female offspring..."

Line 420: It may of course be paternal deposition (though canonically maternal deposition is more prominent), or it may be some other paternal effect. Because dsx is in a sex determination gene, it may be due to paternal imprinting or alternative biological mechanisms other than just CRISPR deposition. A slightly more elaborate discussion on these potential other effects would be appreciated.

Line 473-481: the authors state "Indeed, the pathway to deployment of gene drive mosquitoes recommends that prior to outdoor or open release testing, gene drive-modified mosquitoes are secondarily evaluated in large, overlapping generation indoor cages designed to mimic more closely the native ecological conditions (NASEM, 2016; James et al. 2018)." Caged/confined outdoor releases are not the same thing as an "open release" – the next step following lab studies would likely be confined releases (e.g. caged outdoor/islands/private alleles) and core commitments for these types of releases have been outlined by over 50 members of the field (DOI 10.1126/science.abd1908). I recommend the authors clarify this distinction and these commitments.

I have decided to sign all my reviews to remain scientifically open and honest.

Omar S. Akbari, Ph.D., Associate Professor, Section of Cell and Developmental Biology, University of California, San Diego, La Jolla, California 5101 TATA Hall, La Jolla CA, 92093, (858) 246-0640, oakbari@ucsd.edu

Reviewer #2:

Remarks to the Author:

Gene drives are constructs that can increase in frequency in a population. This research group previously developed a homing type gene drive for population suppression in *Anopheles gambiae* that works by cutting a wild-type allele and then getting copied into the DNA break by homology-directed repair. The drive targeted a highly conserved site (to avoid functional resistance alleles) in a dsx haplosufficient but essential exon for female fertility. It was shown to quickly reach high frequency during cage experiments. At this point, all females would be sterile drive homozygotes, resulting in population suppression.

In this study, the authors scaled up the cage experiments, using larger cage sizes (though adult population size was similar). The large environment and overlapping generations provided a situation closer to that found in wild mosquito populations. The results of the experiment were similar to their early study, eventual suppression of the population. Using modeling, the authors obtain more accurate estimates of some important gene drive parameters. Overall, this study did not contain any particularly new findings, but it does certainly represent an essential step in bringing the gene drive from the lab to the field. The study was conducted generally well, with a few concerns regarding the analysis that can probably be addressed without much trouble.

1. In the introduction, the authors rightly spend some attention on the potential disadvantages of newly released lab lines (an important consideration). However, the actual ability of their large cage system to test this is incomplete. It still involves the use of lab lines, limiting the fitness consequences not directly related to the drive. More importantly, a homing gene drive would quickly transfer to the wild-type genetic background due to its cut-copy-paste mechanism. Thus, any disadvantage of lab lines would likely be transient anyway, causing some delay but not substantially changing the outcome. This should perhaps be mentioned in this section.

2. Is it clear why adult survival increased so much during the large cage experiments, especially when the cages had several weeks before the population suppression experiment started? This change may indicate that something else important is changing in the experiment. Also, it is a little unclear what the statistics in Supplemental Figure 3 refer to. With some modifications to the figure, it should be easier to see what references to male-female differences within each of the six graphs (plus homozygote intersex for the drive survival cages), as well as differences between drive individuals and G3 individuals (this latter comparison should be added if it is not already present).

3. This isn't necessarily an issue, but it might be worth a check: in Figure 2I, the female fertility when both parents are transgenic seems to have roughly the same size error bars as the female fertility when just one parent is transgenic (potentially even a little smaller). However, it is unclear what this means. Presumably, these are females that almost all inherited a drive allele from one parent and a wild-type allele from the other parent. This probably represents only a tiny fraction of the offspring of these crosses because drive conversion is so high and because approximately half of wild-type alleles are converted to R alleles (which would make the female sterile when combined with a drive allele). With this parameter applying to such a tiny number of females, it seems strange that the model gives it only about the same level of error as the other females. Perhaps this is worth a double-check, or an explanation in the text?

4. Line 342: "fertility cost to transgenic females" should be "fertility of transgenic females".

5. As the authors note, somatic, expression, paternal deposition, and maternal deposition are all likely to have approximately similar effects to each other in the cage populations. By allowing them all to vary, any computational analysis would tend to give all three fairly random values, constrained by the net costs for all together, but with large confidence intervals for each. This is exactly what happened. The authors responded by revising their model down to two parameters, fertility for females depending on whether their mother or father was transgenic (a third parameter for fertility if both parents are transgenic is far less important, as noted above). However, they still obtained similar results: high error estimates from two parameters with similar effects, still indicative of overfitting. The mean estimated value of the two parameters was also quite close. I'd suggest therefore that the authors go further. Assume that parental deposition is the same for transgenic mothers and fathers. Then, regardless of whether fitness costs come from parental deposition or somatic costs (or any combination), there will only be a single parameter that needs to be inferred. The confidence interval for such a parameter would probably be fairly tight, allowing the authors to be a little more certain about what is happening in their experimental system. If this new model turns out to have a substantially worse fit, then that would also provide useful information.

6. Lines 370-371 state that, "In spite of the incredible selective pressure exerted by Ag(QFS)1, no mutant alleles were generated that could conceivably code for a functional DSX protein." I certainly agree that this drive did not generate functional resistance alleles at a detectable rate, but I'm also unclear on exactly where this statement comes from. Is it from the next sentence, looking only at three alleles that reached 0.25%? This would make the statement not quite right. Perhaps the authors should revise that statement to say that functional resistance alleles did not prevent suppression of the drive. One could easily imagine a situation where a rate functional resistance allele was generated, but not sequenced and then lost by stochastic effects before it could benefit from its selective advantage. Furthermore, while the three alleles reaching 0.25% caused frameshifts, many more resistant alleles could have been below 0.25%, of which about 1/3 would not have generated frameshift mutations (these were probably still not functional since they did not stop suppression, but again, the " could conceivably code for a functional DSX protein"

doesn't seem quite right when just looking at sequences. It might be better to reword this phrase, or better yet, revise the section to discuss all resistance alleles that formed (eg, reduce the 25% cutoff to whatever is the lowest cutoff that still allows confidence that it was a real sequence), still noting that they did not stop the drive, of course.

7. Consider reversing Supplemental Figure 6 and Figure 2. The former contains the best model, while the latter contains simulations that seem to have a less good fit due to random draws from a parameter space with a fair amount of uncertainty (because of covarying parameters, it might even be better to just leave out the material in Figure 2 and only highly the material in Supplemental Figure 6).

8. Did any of the modeling show a phasing effect, or is this still somewhat unexplained? The discussion may allude to this, but it is hard to see from the figures.

9. In general, the authors try to infer quite a few different parameters from their cages. It is somewhat unclear how much power their method has to infer these, though, even assuming that the model can capture all the dynamics of the caged populations. Though probably not necessarily, the authors could potentially increase their confidence in their inference methods if they demonstrated that it could successfully infer parameters reliably from simulated data.

10. Discussion seems overall a little long. Some edits may be able to make it more concise while keeping all interesting topics, but this is not of high importance.

11. The life history parameter measurement section of the discussion should probably be placed after the large cage experiment sections, so that all these sections are together.

12. It should be noted in the discussion that while the large cage method of the authors represents an improvement over the previous small cages, further important improvements could be made for lab experiments to bring them more closely in line with the natural environment. For example, (and perhaps most importantly), larval competition could be better assessed. In the existing method, all larvae are reared at a density of 200 per tray, then than 400 pupae are added at biweekly intervals. This makes the competition quite indirect and random, but in reality, larva will often compete for limited resources. Gene drive individuals (irrespective of effects on female fertility) may experience small to moderate fitness costs at this critical stage. A large cage method that allowed larva to compete more intensely could better assess if this may be important for gene drive dynamics. This could be particularly important near the end-stages of suppression because the release of competition when the gene drive is at high frequency may affect population dynamics.

We thank both reviewers for their positive appraisals of our work and the recognition of its importance in the pathway of transitioning between lab and field. We are particularly pleased with their constructive suggestions for clarification of some sections regarding the technical detail and context. In addressing these we believe that we have greatly improved the readability of the manuscript and its utility as a reference point.

A detailed list of each of the edits made in response to the reviewers' suggestions is provided below, in line with their original requests.

REVIEWER COMMENTS

Reviewer #1 (Remarks to the Author):

The work by Hammond et. al recapitulates earlier small cage trial experiments (non overlapping discrete generations) in larger cage trials (continuous generations) mimicking conditions in the natural environment, and while they do show invasion/fixation/ and complete population suppression within a year (impressive). They also reveal a much more complicated ecology than was initially observed in earlier trials. Importantly, they did not detect drive-resistant alleles which would be expected as population suppression should exert a strong selection for resistance which would inhibit the spread of the drive. These are the first set of continuous generation gene drive experiments in an environment that mimics natural conditions bringing the technology one step closer to confined field-like conditions. Overall the experiments are well-designed, the manuscript is easy to read and I believe the results are of high importance. I don't have any major criticisms of this work - only have a few minor comments listed below.

Line by line Minor comments:

22: the term "self-sustaining genetic elements" is a very vague term and not necessarily the most precise way to describe GDs in the first sentence.

modified to: "... are genetic elements that bias their inheritance in a self-sustaining manner"

Line 22-24: This sentence is very long, bordering on run on. There are many different ideas tied together here. Consider rephrasing.

Done as suggested.

29: suppressing vs suppression?

Done as suggested.

49: needs a reference

Deredec et al 2011 has been added <https://www.pnas.org/content/108/43/E874.short>

52: If others have adopted progressively rigorous testing protocols, you should cite.

James et al has been added <https://doi.org/10.4269/ajtmh.18-0083>

59: Concept of resistant alleles needs a reference.

We added Burt, A. 2003. Proc Biol Sci 270 (1518): 921–28.
<https://doi.org/10.1098/rspb.2002.2319>.

79: "Since this is what they are designed to emulate" use of 'since' seems non-scientific, consider rephrasing.

Done as suggested.

77-82. Run-on. Rephrase.

We rephrased as following "Indeed, previously developed genetically modified mosquito strains have shown strong fitness costs when tested in large-cage or semi-field experiments that were not observed in initial small cage testing (Aldersley et al. 2019), including severe mating disadvantages that precluded further testing of the strain (Facchinelli et al. 2013). We refer herein to the initiation of these large experiments with the gene drive strain as 'releases', given that this is what they are designed to emulate, albeit they are performed in fully contained chambers that comply with appropriate arthropod containment guidelines."

Line 116: it is not clear how referencing a supplementary figure (S.Fig 1) with a the plasmid map is relevant here when the sentence implies life history traits would be outlined in this figure according to the text.

The reference is correctly pointing to Fig 1 and additional data contained in the supplementary material.

Sup Fig 1: Does the transgene have a terminator for Cas9? If so it should be annotated. Sup Fig 1: Introns annotated with the standard 'pointing up carrot style' would be appreciated in the figure for clarity.

Sup Fig 1: CDS annotated... this is outside the CRISPR cassette, so is this the remainder of Exon 5? If so, a clearer label would be appreciated. If not, then this should be made clearer in the figure as well as legend.

Sup Fig 1: A 2nd chromosome map showing the wt dsx gene map, and gRNA target site would be appreciated for clarity.

We completely revised Supplementary Figure 1 in order to make the features of the transgenic construct and the targeting strategy clearer.

Line 116: A reference for the Kaplan-Meier Survival quantities would be appreciated. Or if not available, a clearer description of what exactly was done here.

We added a description of the Kaplan-Meier survival quantities (with quartiles) used to estimate the average adult population size (pre-release) as requested (Suppl Data 3).

125: How was swarming induced? The prior sentence stated the swarming stimuli was in the cage already. Was it removed and replaced on a regular basis to 'induce' swarming? Or was swarming merely consistently observed in the cages due to permanent inclusion of the swarming stimuli? Clarification here would be appreciated.

Swarming is induced above the swarming visual marker at dusk. Now it is rephrased: "observed".

Figure 1: The Hemotek Feeding system is shown outside of the cage. How was this configured when feeding occurred? Was the whole system inserted? Or were the membranes merely held up to the side of the cage? Or other? Clarification here would be appreciated.

We added the following: "For blood feeding, two Hemotek feeders were introduced in each cage through one of the two openings at the front, leaving the power unit outside."

Sup Fig 2: Inclusion of the temperature and humidity fluctuations were appreciated. I'm

assuming the drastically decreased fluctuations observed between days ~140-230 were due to reduced staff entering and exiting the facility due to quarantine? If so, noting this somewhere in the legend or text would be appreciated, as this event represented a significant change to experimental conditions, and therefore should be noted. If not, and this reduction in ΔT or ΔH was due to some other change in the facility or operating error, it should also be noted.

We extended Supp Fig 2. The plot indicates the daily average data with 1 min interval, which does not show any fluctuation (except 3 points for HR at day ~240, ~270 and ~320). The grey shadings indicated the upper and lower data recorded (still within 1 min interval) indicating the overall change of conditions longer than 1 minute. The relatively decreased min-max data recorded during the period 140-230 days correspond to June-September 2019.

Line 184: you mention that exactly 100 male and female were released into the caged, but then mention they were unsexed? To get these precise numbers, they must have been sexed at some point? Do you simply mean that both males and females (following sexing and counting) were released into the cage simultaneously? If so, please clarify. // Line 186: The AgQFS line does not show sexual dimorphism in pupae, but you released only 100 into cages? To be consistent with the cages described earlier which state 100 male and 100 female were used, should 200 AgQFS have been released? If not, please explain. If other, then this section could use some more explicit clarification.

We specified as following: Sex-specific adult survival was assessed in triplicate for each genotype separately by introducing and sexing 100 male and 100 female pupae of G3 and heterozygous Ag(QFS)1 into either small (0.0049 m³) or large cages (4.7 m³) (Suppl. Fig. 3). In the small cages, we tested 100 individuals in each cage divided by genotype and sex. In each large cage, 100 males and 100 female pupae following sexing and counting were tested together. Because homozygous Ag(QFS)1 do not show clear sex-specific phenotypes as pupae (Kyrou et al. 2018), 100 Ag(QFS)1 total homozygotes (males and 'intersex' females) were introduced into the small and large cages unsexed (Suppl. Fig. 3a).

Figure 2 line 9: "Putatively fertile non-intersex females (ie wt and het)" does this refer to the lines themselves, or the shaded areas? Please specify. Similarly line 11-12 "Red and blue shaded areas indicate fraction of non-intersex females carrying the gene drive in heterozygosity"... these two phrases state basically the same thing "non-intersex females" but refer to very different things on the graph (lines vs shaded areas). Is this a typo? Is one of these supposed to plot the frequency of wt? or true homozygous intersex? In particular the 'shaded' areas of the graph increase drastically just before the population crashes, so this shouldn't probably be referring to non-intersex, but instead true intersex? If not, then please clarify this legend.

We clarified the legend of Fig. 2 as following: Age-structured large (ASL) cages were established over a period of 74 days (shaded grey) and seeded in duplicate with Ag(QFS)1 heterozygous males at low (12.5%, panels a, d, g) and medium (25%, panels b, e, h) allelic frequency, whereas two control cages were maintained without introduction of the Ag(QFS)1 gene drive (c, f). The total egg output (a, b, c), the total frequency of females with apparently normal external morphology (i.e. wild type and heterozygous) (d, e, f), and the frequency of Ag(QFS)1 alleles (g, h) were monitored over time (red and blue lines for replicate cages). Mean egg output of the control is indicated by a dashed line (a, b, c). Red and blue shaded areas indicate the fraction of morphological females that carried the gene drive in heterozygosity (red), or were wild type (blue) (d, e). Arrows indicate the point at which no further eggs were recovered, the point at which populations were considered eliminated. A

total of 20 stochastic simulations of the egg output and the frequency of Ag(QFS)1 (grey lines) were modelled using default parameters based on Kyrou et al. (2018) and expert judgement (Supp. Methods), superimposed to experimental data for the control and gene drive introductions (a, b, c, g, h). The prior and posterior distribution of the relative fertility of Ag(QFS)1 heterozygous females that putatively received deposited nuclease paternally or maternally, as compared to the average fertility of wild-type females (i). Shown in red are the estimates of female fertility from experimental observation in Kyrou et al. (2018).

Line 321: irrespective (not irrespectively). In current usage, it is an adjective describing the genotype

Change as suggested.

Line 346: “Means” is unscientific language. Suggest rephrasing. Perhaps something like “The overlap in the parent-specific estimates therefore precludes determination of parental sex effects on the fertility of transgenic female offspring...”

Revised as following: “By contrast, the marginal posterior distribution for the fertility of females whose mother was transgenic closely resembled its prior distribution, indicating a lack of statistical power to infer this parameter from the ASL data. This reflects the relative rarity of such females, due to both homozygous female sterility and also the heterozygous fitness costs themselves, whereby the cage dynamics are insensitive to their fertility. It is therefore not possible to determine whether or not both types of female offspring differ in fertility on the basis of this data, and it is also not possible to discern the relative roles of parental effects and Cas9 deposition on female fertility. However, the pairwise posterior probabilities of the two parental parameters have negative covariance, indicating that additional information on one parental effect would enable the other to be more accurately determined.”

Line 420: It may of course be paternal deposition (though canonically maternal deposition is more prominent), or it may be some other paternal effect. Because *dsx* is in a sex determination gene, it may be due to paternal imprinting or alternative biological mechanisms other than just CRISPR deposition. A slightly more elaborate discussion on these potential other effects would be appreciated.

We added more elaborate discussion as suggested: “...suggesting a potential role of paternal Cas9 deposition, a phenomenon that we have previously observed in other transgenic strains expressing a nuclease during spermatogenesis (Windbichler et al 2011; Galizi et al. 2016). Additionally we cannot exclude other biological mechanisms that may be related to the position of the gene drive nuclease within a sex determination gene.”

Line 473-481: the authors state “Indeed, the pathway to deployment of gene drive mosquitoes recommends that prior to outdoor or open release testing, gene drive-modified mosquitoes are secondarily evaluated in large, overlapping generation indoor cages designed to mimic more closely the native ecological conditions (NASEM, 2016; James et al. 2018).” Caged/confined outdoor releases are not the same thing as an “open release” – the next step following lab studies would likely be confined releases (e.g. caged outdoor/islands/private alleles) and core commitments for these types of releases have been outlined by over 50 members of the field (DOI 10.1126/science.abd1908). I recommend the authors clarify this distinction and these commitments.

We rephrased as following: “Given their transformative potential, proposed pathways to the deployment of gene drive mosquitoes have been the subject of much discussion recently, yet all recommend a staged, step-by-step pathway, that moves through various levels of

confinement prior to testing in an open release setting (NASEM, 2016; James et al. 2018; Annas et al. 2021; Long et al. 2021). For a gene drive designed to be highly invasive and with a very low threshold of invasion, such as the one described here, its testing in large indoor cages with overlapping generations, designed to mimic designed to mimic more closely the native ecological conditions is a fundamental part of proving its efficacy, in a safe manner (NASEM, 2016; James et al. 2018)."

I have decided to sign all my reviews to remain scientifically open and honest.

Omar S. Akbari, Ph.D., Associate Professor, Section of Cell and Developmental Biology, University of California, San Diego, La Jolla, California 5101 TATA Hall, La Jolla CA, 92093, (858) 246-0640, oakbari@ucsd.edu

Thanks for sharing your comments in a transparent manner!

Reviewer #2 (Remarks to the Author):

Gene drives are constructs that can increase in frequency in a population. This research group previously developed a homing type gene drive for population suppression in *Anopheles gambiae* that works by cutting a wild-type allele and then getting copied into the DNA break by homology-directed repair. The drive targeted a highly conserved site (to avoid functional resistance alleles) in a *dsx* haplosufficient but essential exon for female fertility. It was shown to quickly reach high frequency during cage experiments. At this point, all females would be sterile drive homozygotes, resulting in population suppression.

In this study, the authors scaled up the cage experiments, using larger cage sizes (though adult population size was similar). The large environment and overlapping generations provided a situation closer to that found in wild mosquito populations. The results of the experiment were similar to their early study, eventual suppression of the population. Using modeling, the authors obtain more accurate estimates of some important gene drive parameters. Overall, this study did not contain any particularly new findings, but it does certainly represent an essential step in bringing the gene drive from the lab to the field. The study was conducted generally well, with a few concerns regarding the analysis that can probably be addressed without much trouble.

Thanks for your comments! We will address this in the following.

1. In the introduction, the authors rightly spend some attention on the potential disadvantages of newly released lab lines (an important consideration). However, the actual ability of their large cage system to test this is incomplete. It still involves the use of lab lines, limiting the fitness consequences not directly related to the drive. More importantly, a homing gene drive would quickly transfer to the wild-type genetic background due to its cut-copy paste mechanism. Thus, any disadvantage of lab lines would likely be transient anyway, causing some delay but not substantially changing the outcome. This should perhaps be mentioned in this section.

This study is designed to reveal fitness consequences directly attributable to the drive, rather than differences between lab-adapted and wild-type strains. We released a gene drive into a wild population of the same genetic background in an indoor environment, exactly for the purpose to disentangle the complex contribution of the different backgrounds and to look only at the effect of the transgene. The reviewer is right to point out that the gene drive rapidly sheds most of its original genetic background - this is an attractive feature of gene drives and

the issue of introgression into different genetic backgrounds is very relevant in the optic of field release, but it is out of the scope of this study.

2. Is it clear why adult survival increased so much during the large cage experiments, especially when the cages had several weeks before the population suppression experiment started? This change may indicate that something else important is changing in the experiment. Also, it is a little unclear what the statistics in Supplemental Figure 3 refer to. With some modifications to the figure, it should be easier to see what references to male female differences within each of the six graphs (plus homozygote intersex for the drive survival cages), as well as differences between drive individuals and G3 individuals (this latter comparison should be added if it is not already present).

We extended Suppl Fig3 as suggested and specified the results as following: “No difference in male and female survival were observed for G3 in both small and large cages and for Ag(QFS)1 in small cages. A small difference was observed between Ag(QFS)1 homozygous and heterozygous individuals in both small and large cages before the population experiment.”.

3. This isn't necessarily an issue, but it might be worth a check: in Figure 2I, the female fertility when both parents are transgenic seems to have roughly the same size error bars as the female fertility when just one parent is transgenic (potentially even a little smaller). However, it is unclear what this means. Presumably, these are females that almost all inherited a drive allele from one parent and a wild-type allele from the other parent. This probably represents only a tiny fraction of the offspring of these crosses because drive conversion is so high and because approximately half of wild-type alleles are converted to R alleles (which would make the female sterile when combined with a drive allele). With this parameter applying to such a tiny number of females, it seems strange that the model gives it only about the same level of error as the other females. Perhaps this is worth a double check, or an explanation in the text?

The reviewer is right that only a tiny fraction of mosquitoes will have this phenotype so the “both parents transgenic” parameter isn't very important. We think the confusion here is that these are not inferred parameters, but transformations of inferred parameters; on reflection we had not been clear enough about our method here. The fact the error bar is smaller is because - the model assumes - this cost combines the inferred distribution of paternal and maternal costs, so the overall cost is generally higher than each and therefore close to 1 for most combinations of the sex-specific costs.

On reflection and on the basis of the reviewers comment 5 below, we have decided to revise this figure and no longer present the transformed parameters.

4. Line 342: “fertility cost to transgenic females” should be “fertility of transgenic females”.
Good spot - we have corrected this now.

5. As the authors note, somatic, expression, paternal deposition, and maternal deposition are all likely to have approximately similar effects to each other in the cage populations. By allowing them all to vary, any computational analysis would tend to give all three fairly random values, constrained by the net costs for all together, but with large confidence intervals for each. This is exactly what happened. The authors responded by revising their model down to two parameters, fertility for females depending on whether their mother or father was transgenic (a third parameter for fertility if both parents are transgenic is far less important, as noted above). However, they still obtained similar results: high error estimates

from two parameters with similar effects, still indicative of overfitting. The mean estimated value of the two parameters was also quite close. I'd suggest therefore that the authors go further. Assume that parental deposition is the same for transgenic mothers and fathers. Then, regardless of whether fitness costs come from parental deposition or somatic costs (or any combination), there will only be a single parameter that needs to be inferred. The confidence interval for such a parameter would probably be fairly tight, allowing the authors to be a little more certain about what is happening in their experimental system. If this new model turns out to have a substantially worse fit, then that would also provide useful information.

We thank the reviewer for this thoughtful comment. We think the reviewer has slightly misunderstood what we present in fig. 2I which, on reflection, was not sufficiently clear. They are correct to say that the estimation of three fitness parameters is fairly random and constrained by the overall fitness; what we presented was the overall fitness effects obtained by randomly sampling the posterior distribution and combining the sampled parameters as necessary to obtain the fitness for the given type of female (e.g. a sample for the fitness of heterozygous females with transgenic mother = $(1-x_i)(1-\rho_F)$, where x_i and ρ_F are taken from a sample of the posterior). The parameters shown in Fig. 2I are thus transformations of inferred parameters, rather than inferred parameters themselves, and we can now see that this may cause confusion. The reviewer's suggestion that we reduce the fitness parameters to a single one is interesting, yet one of the aims of the analysis was to explore the covariance of these mechanistic parameters in the posterior, and their relative roles to the dynamics. In this revision, we have simplified the model so that there is a parameter representing the fertility of females with a transgenic father and another for the case of a transgenic mother. (There is no longer a parameter representing additional cost to being transgenic per se - essentially x_i is set to 0 in the previous version of the model). These two fertility parameters, which are now directly inferred, then subsume both parental effects and the role of ectopic expression of Cas9, and are plotted in the new version of fig. 2I. While it may be true that the maternal and paternal costs are the same (our results do not preclude this possibility), we feel that there is a biological rationale for keeping these separate in the model, and showing their joint posterior distribution helps us to discuss the possible causes of the fertility costs. We think this revision is clearer, and we hope the reviewer agrees.

6. Lines 370-371 state that, "In spite of the incredible selective pressure exerted by Ag(QFS)1, no mutant alleles were generated that could conceivably code for a functional DSX protein." I certainly agree that this drive did not generate functional resistance alleles at a detectable rate, but I'm also unclear on exactly where this statement comes from. Is it from the next sentence, looking only at three alleles that reached 0.25%? This would make the statement not quite right. Perhaps the authors should revise that statement to say that functional resistance alleles did not prevent suppression of the drive. One could easily imagine a situation where a rate functional resistance allele was generated, but not sequenced and then lost by stochastic effects before it could benefit from its selective advantage. Furthermore, while the three alleles reaching 0.25% caused frameshifts, many more resistant alleles could have been below 0.25%, of which about 1/3 would not have generated frameshift mutations (these were probably still not functional since they did not stop suppression, but again, the "could conceivably code for a functional DSX protein" doesn't seem quite right when just looking at sequences. It might be better to reword this phrase, or better yet, revise the section to discuss all resistance alleles that formed (eg, reduce the 25% cutoff to whatever is the lowest cutoff that still allows confidence that it was a

real sequence), still noting that they did not stop the drive, of course.

The reviewer is correct to point out that we have not made our logic clear from the main text, as some of the explanation comes from the Figure 3 legend. We chose the cut-off point of 0.25% based on the suggested limit of detection of our sequencing method, from Pfeiffer et al. 2018. All alleles above this threshold of detection were out of frame and thus, not functionally resistant. The figure legend wasn't completely clear that the three mutant alleles were the only mutant alleles detected above the threshold. We have amended the text in the results to make this clearer: "We identified three putative end-joining mutations present above the sequencing threshold frequency of 0.25% (Pfeiffer et al. 2018) in any of the four release cages."

7. Consider reversing Supplemental Figure 6 and Figure 2. The former contains the best model, while the latter contains simulations that seem to have a less good fit due to random draws from a parameter space with a fair amount of uncertainty (because of covarying parameters, it might even be better to just leave out the material in Figure 2 and only highly the material in Supplemental Figure 6).

This is a good suggestion, and we have now reversed those figures. The simulations using parameters drawn from the posterior are now given as the supplementary figure 6; we would like to still include them in the supplementary materials because they demonstrate that the fitted model captures the biology of the cage dynamics, and therefore act as an informal posterior predictive check.

8. Did any of the modeling show a phasing effect, or is this still somewhat unexplained? The discussion may allude to this, but it is hard to see from the figures.

This has been modified accordingly in the main text (results/discussion): "We observed apparent two-weekly fluctuations in the gene-drive frequency time-series, which were confirmed by analysis of autocovariance; we identified positive covariance in all four time-series at a time lag of 2 weeks and negative covariance at shorter time-lags (Supplementary fig. 4). These fluctuations may have resulted from the initial two-phased release in the first week, which would suggest that interbreeding between young and old cohorts of cohabiting adults is rare in the large cage conditions. However, further investigation would be needed to confirm this hypothesis. We note that the fluctuations were not apparent in the model simulated time-series (Supplementary fig. 4). "

9. In general, the authors try to infer quite a few different parameters from their cages. It is somewhat unclear how much power their method has to infer these, though, even assuming that the model can capture all the dynamics of the caged populations. Though probably not necessarily, the authors could potentially increase their confidence in their inference methods if they demonstrated that it could successfully infer parameters reliably from simulated data.

The power to infer model parameters can be assessed by comparing the marginal prior and posterior distributions for those parameters. Our Approximate Bayesian Computation analysis shows that some model parameters are inferable with more accuracy than others from the cage data. For example, we have good power to infer the fertility of females whose father was transgenic, but very little power to infer the fertility of females whose mother was transgenic. In this revision, we have revised the results to clarify this point (2nd para under the heading "Parameter inference reveals drive allele female fertility costs in age-structured mosquito populations"). The reviewer's suggestion is interesting, and would have revealed which parameters are easier to infer in advance of the experiment. However, we prefer to keep the parameters that we included in the model - even when they are hard to infer -

because they are mechanistic parameters with clear biological interpretation, and the differing degrees by which they can be inferred is in itself revealing of their importance to the cage population dynamics. In exception, we have revised the three fertility cost parameters down to two parameters, as discussed in response to pts 3 and 5 above.

10. Discussion seems overall a little long. Some edits may be able to make it more concise while keeping all interesting topics, but this is not of high importance. The life history parameter measurement section of the discussion should probably be placed after the large cage experiment sections, so that all these sections are together.

We revised the discussion, but changing the position of the mentioned section would disrupt our line of argumentation.

12. It should be noted in the discussion that while the large cage method of the authors represents an improvement over the previous small cages, further important improvements could be made for lab experiments to bring them more closely in line with the natural environment. For example, (and perhaps most importantly), larval competition could be better assessed. In the existing method, all larvae are reared at a density of 200 per tray, then than 400 pupae are added at biweekly intervals. This makes the competition quite indirect and random, but in reality, larva will often compete for limited resources. Gene drive individuals (irrespective of effects on female fertility) may experience small to moderate fitness costs at this critical stage. A large cage method that allowed larva to compete more intensely could better assess if this may be important for gene drive dynamics. This could be particularly important near the end-stages of suppression because the release of competition when the gene drive is at high frequency may affect population dynamics.

We agree that larval competition could be important. Further improvements of large cage design should include more realistic conditions for the aquatic life stages. Here, we also chose to focus on those aspects of mosquito biology that would most likely be affected by a disruption of the doublesex gene, including sex specific adult behaviours such as mating and oviposition.

We added to the discussion the following: "In the future, further improvements to the cage design could be made, such as establishing more realistic conditions for aquatic life stages that more faithfully recapitulate larval competition. Nonetheless, the Ag(QFS)1 strain is the first gene drive strain to pass this essential intermediate step..."

Reviewers' Comments:

Reviewer #1:

Remarks to the Author:

The authors have sufficiently address all of my suggestions.

Reviewer #2:

Remarks to the Author:

The authors have done a good job with the revisions, and the paper is now ready for publication. I don't have any new comments, so I'll just respond to the authors' responses below (as can be seen, the issues were largely resolved satisfactorily). These are just residual comments that need not delay acceptance of the manuscript.

1. I must have imagined seeing a discussion of genetic differences between the release strain and natural strains and was actually trying to make the same point as the authors in their response (that it was not directly relevant to the cage experiments). Apologies for the confusion.

2. Figure S3 looks nice now. Please see the first part of my comment, though. Why did everyone have much longer lifespans at the end of the large cage experiment compared to before it? Since this experiment is dedicated to assessing more complex life histories of the mosquitoes in more natural conditions, this sort of thing can be potentially important to any conclusions. I understand that it might be difficult to figure this out, but I think it deserves at least an additional sentence of two of discussion.

3. Okay, I understand this now. I thought that there were two separate models with different numbers of parameters (with the three-parameter fitness model displayed in 2I). The new figure of course clears this up.

4. This problem was fixed.

5. I think this is also successfully cleared up in the revisions and is much clearer now. However, while a 2-parameter model with fitness dependent on sex of the drive parent certainly has biological relevance (representing differences in paternal and maternal deposition with potential somatic deposition added on top to each case equally), I think that a 1-parameter model is equally relevant (representing no significant parental deposition or perhaps equal between sexes, potentially with somatic effects alone being the only substantial factor in reducing fertility). If the authors don't perform this analysis, it should be at least mentioned as a possibility.

6. These revisions are fine - it just needed a wording change.

7. Good change.

8. This is interesting. I like the changes, and perhaps the authors can go a little further in terms of interpreting their data. This is just a suggestion for if the authors think they can squeeze some more insight into real populations out of the "phasing" observation. Since females are likely to only mate once, perhaps this indicates that older male lose their ability to effectively compete for mates? In this case, it would be more important to release younger drive males in the field.

9. Seems reasonable.

10-11. No problems here.

12. Good change.

Minor note: the authors declare no conflicts of interest in the acknowledgements, but then some authors have a conflict of interest in the "Declaration" section.

We thank both reviewers for their positive appraisals of our work and the recognition of its importance in the pathway of transitioning between lab and field. Their suggestions greatly improved the quality of our manuscript and its utility as a reference point.

A detailed list of each of the edits made in response to the reviewers' suggestions is provided below, in line with their original requests.

REVIEWER COMMENTS

Reviewer #1 (Remarks to the Author):

The authors have sufficiently address all of my suggestions.
Thank you for your constructive suggestions.

Reviewer #2 (Remarks to the Author):

The authors have done a good job with the revisions, and the paper is now ready for publication. I don't have any new comments, so I'll just respond to the authors' responses below (as can be seen, the issues were largely resolved satisfactorily). These are just residual comments that need not delay acceptance of the manuscript.

Thank you for your constructive suggestions.

1. I must have imagined seeing a discussion of genetic differences between the release strain and natural strains and was actually trying to make the same point as the authors in their response (that it was not directly relevant to the cage experiments). Apologies for the confusion.

2. Figure S3 looks nice now. Please see the first part of my comment, though. Why did everyone have much longer lifespans at the end of the large cage experiment compared to before it? Since this experiment is dedicated to assessing more complex life histories of the mosquitoes in more natural conditions, this sort of thing can be potentially important to any conclusions. I understand that it might be difficult to figure this out, but I think it deserves at least an additional sentence of two of discussion.

In the Results we say (line 361-362): "*In agreement with Pollegioni et al. (2020), suggesting the difference is due to the micro-environmental conditions of the large cages and not due to strain adaptation or the genotypes*".

3. Okay, I understand this now. I thought that there were two separate models with different numbers of parameters (with the three-parameter fitness model displayed in 2I). The new figure of course clears this up.

We are pleased that the reviewer finds this clearer now.

4. This problem was fixed.

5. I think this is also successfully cleared up in the revisions and is much clearer now. However, while a 2-parameter model with fitness dependent on sex of the drive parent certainly has biological relevance (representing differences in paternal and maternal deposition with potential somatic deposition added on top to each case equally), I think that a 1-parameter model is equally relevant (representing no significant parental deposition or perhaps equal between sexes, potentially with somatic effects alone being the only substantial factor in reducing fertility). If the authors don't perform this analysis, it should be at

least mentioned as a possibility.

We are pleased that the reviewer finds the revision is now clear on this point. Though we didn't separately fit a single fitness parameter model, we feel this is unnecessary because the single parameter version is a limiting case of the two parameter model that we did fit. However, we did discuss the possibility that the two fitness costs have a common cause (ectopic activity of Cas9) and are the same. In the Results we say *"It is therefore not possible to determine whether or not both types of female offspring differ in fertility on the basis of this data, and it is also not possible to discern the relative roles of parental effects and Cas9 deposition on female fertility"*, and in the Discussion we say *"We were unable to identify the potential causes of fertility cost from our analysis of the large cage observations"*. We hope these statements (and the text around them) make it clear that we do not rule out the possibility that the costs are the same.

6. These revisions are fine - it just needed a wording change.

7. Good change.

8. This is interesting. I like the changes, and perhaps the authors can go a little further in terms of interpreting their data. This is just a suggestion for if the authors think they can squeeze some more insight into real populations out of the "phasing" observation. Since females are likely to only mate once, perhaps this indicates that older male lose their ability to effectively compete for mates? In this case, it would be more important to release younger drive males in the field.

9. Seems reasonable.

10-11. No problems here.

12. Good change.

Minor note: the authors declare no conflicts of interest in the acknowledgements, but then some authors have a conflict of interest in the "Declaration" section.

Deleted from the section acknowledgments. Thanks for spotting this.